# The Effect of Physical Pain on Depression and Resilience: A Cross-Sectional Study

**DOI:** 10.3390/healthcare14010053

**Published:** 2025-12-25

**Authors:** Rubén Fernández-García, Gonzalo Granero-Heredia, Maria Rosa Ortega-Lasheras

**Affiliations:** Department of Nursing, Physiotherapy and Medicine, University of Almeria, 04120 Almeria, Spain

**Keywords:** physical pain, depression, health, resilience

## Abstract

**Objectives**: The main objective of this study was to assess the relationship between physical pain, depression, and resilience in a convenient group of university students. **Methods**: A comparative, descriptive, and exploratory study was carried out. The sample comprised 2305 university students enrolled in the degrees of physiotherapy, nursing, medicine, and physical activity and sport sciences. The ‘chronic pain assessment questionnaire’, the ‘brief resilience scale’ and the ‘depression, anxiety and stress scale’ were used. **Results**: The results indicated that the model had an Incremental Fit Index = 0.94, a Comparative Fit Index = 0.93, a Normalised Fit Index = 0.91, and a Root Mean Squared Error of Approximation = 0.045. The value of X^2^(5) = 12.35 with *p* < 0.05 was also reported. These data support the validity of the theoretical model developed. The results indicate that physical pain has a negative effect on depression (β = 0.55, *p* < 0.001). Furthermore, statistically significant negative associations were found between pain and resilience (β = −0.40, *p* < 0.002). Finally, a negative relationship between resilience and depression was also observed (β = −0.35, *p* < 0.0039). **Conclusions**: New strategies and therapies need to be developed to improve the quality of life of patients with chronic pain.

## 1. Introduction

Overall, 25.9% of the adult population in Spain suffers from chronic pain (CP). It is more prevalent in women (30.5%) than in men (21.3%) [1].

CP does not tend to be an isolated problem and is often associated with other signs and symptoms. This means that it is often accompanied by other health problems such as fibromyalgia, major depressive disorder, or generalised anxiety. More than 67% of patients with CP also have a mental health problem [2,3]. In terms of the location of CP, 11.1% indicate that it is located in a specific area of the body. In 10.1% of cases, it is located in the back, while 7.1% of patients have pain in their legs and feet. Pain in the arms and hands affects 4.1% of patients, and in 3.5% of cases, the pain is experienced in the head. CP is also associated with metabolic disturbances and cognitive impairment [4]. Between 5% and 14% of patients have attempted suicide at some point and around 20% have suicidal ideation. These figures are a clear indication of the degree of suffering experienced by people with CP. Moreover, a significant number of patients die from opioid overdose [5,6]. Considering the diversity of pathologies related to CP, it can be concluded that there is no single physiological finding that can encompass all the factors that contribute to the symptoms of CP [7].

Different measurement instruments are available for assessing CP. The Brief Pain Inventory (BPI) is used to assess perceptions of pain and its impact on daily life [8]. There is also the McGill Pain Questionnaire (SF-MPQ-2), which includes a picture of pain location, previous analgesic use, and experiences of pain history [8]. Depression screening is recommended for patients suffering from CP. The Minnesota Multiphasic Personality Inventory II (MMPI-2) and the Beck Depression Scale are two good measurement instruments which are frequently used in patients with CP. Current treatments for CP can reduce a patient’s pain by approximately 30%, significantly improving their function and quality of life.

Figures estimate that approximately 264 million people worldwide suffer from depression, a mental illness that both children and adults can experience [9]. Nearly 14% of adolescents will meet criteria for depression before the age of 18, and approximately 25% have experienced a depressive episode. The rate of depression in this age group is expected to increase progressively over time to an estimated 34% [10]. This situation poses a major challenge for the healthcare system. Depression in adolescents often leads to severe social and educational impairment, as well as a high rate of substance use, other associated mental disorders, dissatisfaction with life, and even suicide risk [11].

A meta-analysis conducted in 2025 analysed the prevalence of mental disorder symptoms in university students worldwide. A systematic search of seven databases was conducted according to the PRISMA guidelines. A total of 1655 primary studies from 62 meta-analyses involving 8,706,185 participants were included. Regarding data on depression, the study indicated that 35.4% suffered mild depression symptoms and approximately 13.4% presented symptoms of severe depression [12]. Studies indicate that depression and anxiety are the two most prevalent psychological problems in the university community. In Spain, the prevalence of depression ranges from 13.5 to 18.4%. Cases of anxiety range from 23.6 to 44.7%. The mental health of university students appears to have deteriorated in recent years due to global stressors [13,14]. It is therefore essential to prepare health professionals so that they are able to manage suffering and frustration. Thus, it is important to build resilience, which is the ability to bounce back and become stronger after failure and be positive in the face of problems or adversity. Resilience is not just a personality trait, but a capacity that can be enhanced and developed over time [15].

This study focuses on resilience in the face of pain. It is a variable that largely depends on the individual person. It is the ability to carry out activities of daily living despite being in pain, as well as the ability to manage one’s emotions and thoughts [16]. The highest levels of pain resilience have been found in laboratory tests, while the lowest levels were found in cross-sectional studies and longitudinal research after a 3-month interval [17]. Neuroimaging studies have identified changes in brain volume and psychophysiological correlation in people who are more resilient to pain [18].

Our research hypotheses are as follows: physical pain has a positive effect on depression, physical pain has a negative effect on resilience, and resilience is a mediating variable of the depressive state. The aim of this study was to analyse the relationship between pain, depression, and resilience in university students studying health-related fields.

## 2. Materials and Methods

### 2.1. Design and Participants

In the present study, the steps outlined in the STROBE statement [19] were used as a reference. The study was exploratory, descriptive, cross-sectional, and comparative. The sample comprised 2305 university students enrolled in the degrees of physiotherapy, nursing, medicine, and physical activity and sport sciences, of which 45 were excluded for not responding adequately to the items proposed. The final sample consisted of 2305 students from various universities located in the north, south, and east of Spain. Some of these include the University of Oviedo, the University of Almería, and the University of Granada. Non-probability and convenience sampling were used. The sampling error of this study was 0.027, with a confidence interval of 95.0%.

The inclusion criteria were being a university student and being over 18 years of age. No exclusion criteria could affect the analyses.

### 2.2. Instruments and Variables

The Spanish adapted version of the Chronic Pain Assessment Questionnaire was used. The questionnaire comprises eight items that measure two dimensions: pain intensity (3 items), measured using a VAS scale (0–10), and functional disability (8 items) in performing social, work, and leisure activities [20]. Cronbach’s alpha (α = 0.82). The McDonald’s omega was ω = 0.86.

The Spanish adapted version of the Depression, Anxiety and Stress Scale (DASS-21) was used [21]. This instrument consists of a total of 21 items that are assessed using a Likert scale. It has the advantage of being self-administered, easy and simple to answer, and has reported excellent psychometric properties in the general population, in adolescents, and in university students. In the reliability analysis, the questionnaire obtained an overall value of α = 0.917. For Depression, Anxiety and Stress, it had values of α = 0.903, α = 0.890 and α = 0.915, respectively. On this occasion, we have only used the items corresponding to the DASS-depression subscale.

The Spanish and adapted version of the Brief Scale of Resilience with only 10 items was used in this study [22]. Each item was scored on a 5-point Likert-type scale (4: almost never true; 3: rarely true; 2: sometimes true; 1: often true; and 0: almost always true). In terms of psychometric properties, the test has unidimensional internal structure validity [Comparative Fit Index (CFI) = 0.97, Root Mean Squared Error of Approximation (RMSEA) = 0.05, Standardised Root Mean-Square (SRMR) = 0.03] and adequate internal consistency reliability (α = 0.85).

### 2.3. Procedure

Considering the instruments necessary to carry out the research, Google Forms was used to create a questionnaire with all the items of interest. The university students who met the inclusion criteria completed the questionnaire through an individual link that had been sent to them via the University of Almería’s corporate email. To ensure that the questionnaire was answered correctly, three items were duplicated. In cases where the answers to these questions did not match, the subject’s participation was disregarded, which resulted in forty-five responses being eliminated. Data were collected between January and June 2025. All participants participated on a voluntary basis after giving informed consent. The ethical criteria established in the Declaration of Helsinki were followed. In addition, the study was approved by the ethics committee of the University of Almeria (EFM 419.25).

### 2.4. Data Analysis

The IBM SPSS V29.02 statistical package was used for the statistical analysis of the results. Firstly, the normality of the results was analysed by means of the values of skewness and kurtosis of each item, for which values of less than 2 were obtained. Likewise, the reliability of the instruments used was assessed through Cronbach’s Alpha and McDonald’s Omega tests, establishing the reliability index at 95%.

The IBM AMOS v29.0 statistical package was used to build the structural equation model. To fit the model, the values of the Incremental Fit Index (IFI), Comparative Fit Index (CFI), together with the Normalised Fit Index (NFI) were consulted. The values of these indices must be greater than 0.90 to show a good fit. The RMSEA was also used. The values of this index must be less than 0.08 [23].

Figure 1 presents the theoretical model showing the direction of the causal relationships of the variables. The theoretical model is made up of 2 endogenous variables and 1 exogenous variable. Unidirectional causal relationships were observed. These occur when one-way arrows appear, where the origin of the arrow is the independent variable and the tip is the dependent variable. Figure 1 shows that physical pain is a mediating variable that affects resilience and depression. On the other hand, resilience affects depression, under the mediating effect of physical pain.

## 3. Results

Table 1 shows the mean values, standard deviations, skewness, and kurtosis for each of the variables. It also presents the correlational matrix of the variables that make up the structural equation model. The correlational analysis shows significant positive and negative correlations between all the variables that make up the study. We found a negative correlation between physical pain and resilience (r = −0.42; *p* < 0.01). A positive correlation was found between physical pain and depression (r = 0.58; *p* < 0.01). Finally, a negative correlation was observed between resilience and depression (r = −0.45; *p* < 0.01). Taking into account the results, the data show a normal distribution [24].

The results indicated that the proposed model presented a good fit to the data: IFI = 0.94, CFI = 0.93, NFI = 0.91 and RMSEA = 0.045. The value of X^2^(5) = 12.35 with *p* < 0.05.

These data support the validity of the theoretical model developed. The results (Table 2) show that the variable of physical pain (PP) had an effect on depression (D) (β = 0.55, *p* < 0.05). The findings also demonstrate a correlation, whereby the greater the perception of PP, the worse the depressive state. Furthermore, statistically significant negative associations were found between physical pain and resilience (R) (β = −0.40, *p* < 0.05). It can be interpreted that the higher the level of pain, the lower the capacity to cope with conflict situations.

Finally, a negative effect was found between resilience and depression (β = −0.35, *p* < 0.05), with physical pain negatively modulating resilience. This result does not allow us to comment on the importance of resilience in the control of depressive states.

## 4. Discussion

University education in the field of health is known for its demanding curriculum, heavy workloads, complex decision-making, emotional challenges, clinical rotations, and professional placements. As a result, anxiety and depression problems may occur in this type of university student. There is a great deal of research focused on the study of resilience as a protective variable against psychological distress in university students [25,26]. Related to the topic of this research, a recently published study evaluated the effect of variables such as anxiety, depression and professional burnout on symptoms such as back, neck, and head pain in medical students, identifying statistically significant differences [27].

The results of this study indicate that the physical pain variable directly affects both depression and resilience, which is in line with the existing literature. Numerous studies associate an increase in pain symptoms with more severe depressive states [28]. The positive relationship found between the two variables suggests that physical pain not only affects functional aspects or the inability to perform everyday tasks, but also psychological well-being, thus leading to suffering. Moreover, the longer the duration of pain, the greater the possibility of experiencing emotional suffering [29].

It is also pertinent to acknowledge the negative relationship between the variables of resilience and physical pain. Resilience is sometimes described as the ability to cope with adversity [30]. In this regard, it is understood that the more resilient a person is, the more strategies they have to cope with events or situations that cause anxiety and discomfort. Some studies characterise resilience as the result of a process of adaptation to a range of situations that produce discomfort and stress. This adaptive capacity to develop internal resources to cope with conflictive and anxious situations is measured by the subject’s level of personal growth and well-being. Therefore, it is highly likely that people who are less able to cope with major life problems are more likely to experience physical pain. This could be due to poor coping skills, which are a result of not knowing or not having learned effective ways to manage their emotional states [31].

In contrast, statistically significant negative associations were found between resilience and depression. This result may indicate the protective effect of resilience on depressive symptoms, despite the modulating effect of pain. The literature supports these findings, suggesting that resilience enables people to view their situation in a more positive and optimistic way. Therefore, it is possible to improve the psychological component of pain and develop more strategies or techniques for better emotional management [32]. Related to this research, studies indicate that psychological interventions in patients with physical pain can reduce levels of catastrophic thinking and improve psychological resilience. Catastrophic thoughts commonly occur in patients with depression [33].

Considering the results obtained, there are many variables that can affect the experience of pain, including psychological, physical, and environmental factors. It is essential to work with or help patients with CP to manage their feelings and emotions.

Social support is a highly relevant variable in the relationships between pain, resilience, and depression. Evidence shows that people with a strong and firm support network experience lower levels of depression and a better quality of life [34]. Social support also plays a crucial role in the relationship between pain, resilience, and depression. Having social support is perhaps one of the most important factors associated with good health and quality of life [35]. It is of utmost importance to have the support of the people who love and respect us, given that human beings need the affection and appreciation of others. Above all, we need to feel respected and recognised by people with whom we have a strong emotional bond. In this regard, it has been shown that people with strong support networks experience lower levels of depression and a better quality of life compared to those who cope with pain in isolation [36].

## 5. Limitations and Future Perspectives

This study has various limitations to be considered. Firstly, it would be of great interest to develop a longitudinal study to be able to examine the directionality of the variables. Moreover, future research could analyse new variables alongside some of the variables used in this study or different variables could be used all together. For example, the importance of social support or access to health services could be assessed. Although the sample is representative, it may not fully reflect the diversity of the CP population. Further studies should include larger and more heterogeneous samples. Another limitation is related to the use of self-reported measures, which may be subject to social desirability bias or recall errors. Including more controllable measurement tools could provide a more objective assessment. Finally, we should acknowledge that only students studying in the field of health were included. To further research on this topic, including university students from other areas of knowledge could be considered in future studies.

## 6. Conclusions

This study reflects the complex relationship between PP, resilience, and depression. The results indicate that CP has a direct impact on physical and emotional health. It exacerbates states of depression and decreases people’s ability to cope. Resilience is a key variable that can be protective in the ability to manage CP. It is vital to adopt a comprehensive approach to CP management, addressing its many causes and predisposing factors. It is also crucial to develop new intervention strategies to improve health and quality of life in patients with CP.

These findings emphasise the importance of adopting a multidimensional approach to the care of patients with CP, integrating psychological strategies that promote resilience and emotional well-being. Future studies could continue to explore interventions that optimise the quality of life of affected individuals, thus contributing to the development of more effective and accessible therapeutic strategies.

## Figures and Tables

**Figure 1 healthcare-14-00053-f001:**
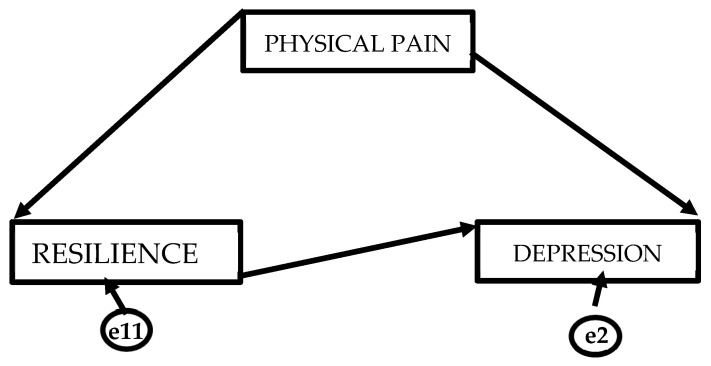
Theoretical representation of the structural equation model.

**Table 1 healthcare-14-00053-t001:** Descriptive reliability and correlational analyses of the study variables.

	M	SD	SKEW	KUR	2. R	3. D
1. PP	2.10	1.30	0.90	1.20	−0.42 **	0.58 **
2. R	3.00	0.80	−0.50	0.20		−0.45 **
3. D	1.20	0.75	0.80	1.10		

Note: ** *p* < 0.01. PP: physical pain; D: depression; R: resilience.

**Table 2 healthcare-14-00053-t002:** Standardised causal relationship of the variables.

Direction Causal Relationships	Estimation Error	Regression Weights	Standardised Weights	
		Critical Ratio	*p*	β
PP → D	0.12	4.58	0.01	0.55
PP → R	0.10	−4.00	0.02	−0.40
PP → R + D	0.09	−3.80	0.039	−0.35

PP: physical pain; D: depression; R: resilience.

## Data Availability

The data used to support the findings of the current study are available from the corresponding author upon request.

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
