# Peer review of "Healthcare2026, 14(1), 53;https://doi.org/10.3390/healthcare14010053"

_healthcare, 2025, doi:10.3390/healthcare14010053_

Round 1

Reviewer 1 Report

Comments and Suggestions for Authors

This study evaluates the relationship between physical pain, depression, and resilience in a large sample of Spanish university students (n=2305) using self-report questionnaires and a structural equation model (SEM). It reports that pain is positively associated with depression and negatively associated with resilience, and that resilience is negatively associated with depression. The topic is relevant, and the sample size is a strength. However, the manuscript presents significant problems that must be addressed before publication:

Major Comments
1. The study refers to “patients with chronic pain,” but it must be demonstrated that chronicity (duration ≥3 months, etc.) and/or chronic pain criteria were measured.

2. The “Chronic Pain Assessment Questionnaire” (8 items) is mentioned, and a Spanish adaptation related to grading chronic pain is cited, but it is unclear which exact scale was applied, how it is scored, what ranges it has, or whether it measures intensity, interference, and/or chronicity.

3. Regarding resilience, it is referred to as a “Brief Scale of Resilience with 10 items,” but the cited reference corresponds to the CD-RISC 10 (Connor-Davidson), and furthermore, the response description (0–4 with “never…always” labels) does not align with the standard presentation of the BRS/CD-RISC in scientific evidence.

4. Hypothesis 3 states: “Resilience is a modulating variable of the depressive state,” which suggests moderation. However, the reported SEM includes direct pathways: pain→depression, pain→resilience, and resilience→depression, which corresponds to a partial mediation model (or direct effects), not moderation. Additionally, Table 2 shows a pathway “PP→R+D” that is not clearly defined and suggests an incorrect formulation of the model. Please clarify this.

5. It is mentioned that a "Multigroup equation model" will be performed, but no multigroup analysis is reported.

6. The results section states "positive and significant correlations between all variables," but then indicates a negative correlation between pain and resilience, and between resilience and depression.

7. The sign of some correlations does not match the text (e.g., "0.45**" appears where the text reports r = -0.45 between resilience and depression).

8. Table 2 shows "Regression Weights / Standardized Weights," but essential elements are missing: unstandardized coefficients (B), standard error (SE), CR, and p are presented, but without a clear estimator or units; furthermore, the heading "Critical Radius" is incorrect.

9. Essential descriptive variables are not reported (at least in the provided text): mean age and range, distribution by sex, degree, university, year of study, etc.

Minor comments
1. “Social support” is extensively included in the discussion, but it was not measured in the study.
2. Reference number 15 is missing (it jumps from 14 to 16).

Author Response

Dear reviewer,

I appreciate your recommendations for publishing a higher quality paper.  I will highlight the changes you indicate in purple. Thank you.

This study evaluates the relationship between physical pain, depression, and resilience in a large sample of Spanish university students (n=2305) using self-report questionnaires and a structural equation model (SEM). It reports that pain is positively associated with depression and negatively associated with resilience, and that resilience is negatively associated with depression. The topic is relevant, and the sample size is a strength. However, the manuscript presents significant problems that must be addressed before publication:

Major Comments

  1. The study refers to “patients with chronic pain,” but it must be demonstrated that chronicity (duration ≥3 months, etc.) and/or chronic pain criteria were measured.

I am sorry. I think I did not explain it correctly. The participants are university students over the age of 18, not patients with chronic pain.

  1. The “Chronic Pain Assessment Questionnaire” (8 items) is mentioned, and a Spanish adaptation related to grading chronic pain is cited, but it is unclear which exact scale was applied, how it is scored, what ranges it has, or whether it measures intensity, interference, and/or chronicity.

Solved.. Line 96-99

  1. Regarding resilience, it is referred to as a “Brief Scale of Resilience with 10 items,” but the cited reference corresponds to the CD-RISC 10 (Connor-Davidson), and furthermore, the response description (0–4 with “never…always” labels) does not align with the standard presentation of the BRS/CD-RISC in scientific evidence.

Solved. I am sorry for the mistake. Line 109-111

The cite reference corresponds to Notario-Pacheco B, Solera-Martínez M, Serrano-Parra MD, Bartolomé-Gutiérrez R, García-Campayo J, Martínez-Vizcaíno V. Reliability and validity of the Spanish version of the 10-item Connor-Davidson Resilience Scale (10-item CD-RISC) in young adults. Health Qual Life Outcomes. 2011;9:63. doi:10.1186/1477-7525-9-63

  1. Hypothesis 3 states: “Resilience is a modulating variable of the depressive state,” which suggests moderation. However, the reported SEM includes direct pathways: pain→depression, pain→resilience, and resilience→depression, which corresponds to a partial mediation model (or direct effects), not moderation. Additionally, Table 2 shows a pathway “PP→R+D” that is not clearly defined and suggests an incorrect formulation of the model. Please clarify this.

Dear reviewer. Perhaps I did not use the correct word; resilience is not a modulating variable of depression, it is a mediating variable. On the other hand, Table 2 shows a pathway ‘PP→R+D’. I mean that resilience is a mediating variable of depression, but under the mediating effect of physical pain.

  1. It is mentioned that a "Multigroup equation model" will be performed, but no multigroup analysis is reported.

I am sorry for the mistake. It has already been resolved. Line 134

  1. The results section states "positive and significant correlations between all variables," but then indicates a negative correlation between pain and resilience, and between resilience and depression.

Solved Line 159

  1. The sign of some correlations does not match the text (e.g., "0.45**" appears where the text reports r = -0.45 between resilience and depression).

Solved

  1. Table 2 shows "Regression Weights / Standardized Weights," but essential elements are missing: unstandardized coefficients (B), standard error (SE), CR, and p are presented, but without a clear estimator or units; furthermore, the heading "Critical Radius" is incorrect.

I am sorry, I do not understand what you mean when you say:,”but without a clear estimator or units”

  1. Essential descriptive variables are not reported (at least in the provided text): mean age and range, distribution by sex, degree, university, year of study, etc.

Dear reviewer, as structural equation modelling was used, the sociodemographic variables recorded were not taken into account. In previous studies published in the Healthcare journal, it was not necessary to include sociodemographic variables. Your proposal is very interesting, but in order to facilitate the publication of the article as soon as possible, we have decided not to include the variables you mention. Thank you. 

Minor comments

  1. “Social support” is extensively included in the discussion, but it was not measured in the study.

Dear reviewer, you are right, but I wanted to use the end of the discussion to talk about social support, due to its mediating effect on a large number of psychological variables.

  1. Reference number 15 is missing (it jumps from 14 to 16).

Solved

Best regards

Reviewer 2 Report

Comments and Suggestions for Authors

Manuscript Title: The Effect of Physical Pain on Depression and Resilience: a cross-sectional study

Overall Assessment:

This study investigates the structural relationships between physical pain, depression, and resilience among university students (specifically in health sciences). The use of a large sample size and the application of Structural Equation Modeling (SEM)  are the study's strongest points. The hypothesized model (Pain -> Depression and Resilience) fits the data well. However, the manuscript suffers from carelessness in writing, table presentation, and a lack of depth in the discussion. Specifically, the outdated references (many from the early 2000s) and the lack of contextual integration in the discussion need to be addressed.

Abstract

  • Presentation of Results: The abstract provides SEM fit indices), which is excellent. However, the formatting of p-values is inconsistent. Standardizing this notation is necessary.
  • Language/Typos: There are typographical errors such as "Depressión" (Spanish accent mark used in English text)  which should be corrected.
  1. Introduction
  • Rationale: The introduction summarizes the link between chronic pain and depression well. However, the rationale for selecting health science students (physiotherapy, nursing, medicine) as the sample is not sufficiently justified. Are these students treated as "future health professionals" with specific stressors, or simply as a convenient group of "university students"? This distinction should be clarified to contextualize the findings.
  • Hypotheses: Hypothesis 3 states: "Resilience is a modulating variable of the depressive state". However, the path diagram (Figure 1) clearly depicts resilience as a mediator between pain and depression (Pain -> Resilience -> Depression) . In statistics, "modulation" often implies moderation (interaction), whereas the model tests mediation. The terminology should be corrected to "Mediating variable" to align with the analysis.
  1. Materials and Methods
  • Sample: The reporting of the sample size and sampling error (0.027) is transparent and strong.
  • Procedure: Data collection via "Google Forms" is mentioned. The authors should briefly clarify how they ensured that only the target population filled out the form (e.g., distribution via university emails). The exclusion of 45 participants due to inconsistent answers on duplicated items is a good quality control measure .
  1. Results
  • Table 1 : In the header of Table 1, the column labeled "SWEK" is likely a typo for "SKEW" (Skewness). This looks unprofessional and must be fixed.
  • Table 2 : In Table 2, under the column headings, it reads "Critical Radio". The correct statistical term is "Critical Ratio" (C.R.). Additionally, the use of commas vs. dots for decimals in p-values is inconsistent.
  • Model Interpretation: The SEM results (Figure 1 and Table 2) indicate that physical pain directly increases depression (beta=0.55) and decreases resilience (beta=-0.40), while resilience decreases depression (beta=-0.35). These findings are consistent with the literature and statistically robust.
  1. Discussion
  • Depth: The discussion relates findings to the literature but remains somewhat superficial. It fails to discuss the specific context of health science students—such as academic stress, clinical placement anxiety, or future career concerns—and how these factors might interact with pain and depression.
  • References: The reference list contains a significant number of older sources (early 2000s) . Since this is a current topic, incorporating more research from the last 5-10 years would enhance the manuscript's relevance.

Conclusion and Recommendation

The study is worthy of publication due to its large sample size and robust statistical analysis (SEM). However, the typographical errors in tables (Critical Radio, SWEK) and text (depressión) give the impression of a hastily prepared manuscript. These errors must be corrected, and the discussion should be deepened to reflect the specific sample characteristics.

Summary of Required Revisions:

  1. Types: Correct "SWEK" to "SKEW" (Table 1), "Critical Radio" to "Critical Ratio" (Table 2), and "depressión" to "depression" (Text).
  2. Terminology: Clarify Hypothesis 3; the model suggests "Mediation," but the text says "Modulating."
  3. Sample Justification: Explain why health science students were chosen and discuss the implications of this specific group.

Author Response

Reviewer

I appreciate your recommendations for publishing a higher quality paper.  I will highlight the changes you indicate in red. Thank you.

Overall Assessment:

This study investigates the structural relationships between physical pain, depression, and resilience among university students (specifically in health sciences). The use of a large sample size and the application of Structural Equation Modeling (SEM)  are the study's strongest points. The hypothesized model (Pain -> Depression and Resilience) fits the data well. However, the manuscript suffers from carelessness in writing, table presentation, and a lack of depth in the discussion. Specifically, the outdated references (many from the early 2000s) and the lack of contextual integration in the discussion need to be addressed.

Abstract

  • Presentation of Results:The abstract provides SEM fit indices), which is excellent. However, the formatting of p-values is inconsistent. Standardizing this notation is necessary.

Solved Line 17-19

  • Language/Typos:There are typographical errors such as "Depressión" (Spanish accent mark used in English text)  which should be corrected.

Solved.

  1. Introduction
  • Rationale:The introduction summarizes the link between chronic pain and depression well. However, the rationale for selecting health science students (physiotherapy, nursing, medicine) as the sample is not sufficiently justified. Are these students treated as "future health professionals" with specific stressors, or simply as a convenient group of "university students"? This distinction should be clarified to contextualize the findings.

They are a convenient group of university students to examine, as indicated in the title, the relationship between physical pain, depression, and resilience. Line 9

  • Hypotheses:Hypothesis 3 states: "Resilience is a modulating variable of the depressive state". However, the path diagram (Figure 1) clearly depicts resilience as a mediator between pain and depression (Pain -> Resilience -> Depression) . In statistics, "modulation" often implies moderation (interaction), whereas the model tests mediation. The terminology should be corrected to "Mediating variable" to align with the analysis.

Solved. Line 79

  1. Materials and Methods
  • Sample:The reporting of the sample size and sampling error (0.027) is transparent and strong.
  • Procedure:Data collection via "Google Forms" is mentioned. The authors should briefly clarify how they ensured that only the target population filled out the form (e.g., distribution via university emails). The exclusion of 45 participants due to inconsistent answers on duplicated items is a good quality control measure .

The university students who met the inclusion criteria completed the questionnaire through an individual link that had been sent to them via the University of Almería's corporate email. Line 119

  1. Results
  • Table 1 :In the header of Table 1, the column labeled "SWEK" is likely a typo for "SKEW" (Skewness). This looks unprofessional and must be fixed.

Solved

  • Table 2 :In Table 2, under the column headings, it reads "Critical Radio". The correct statistical term is "Critical Ratio" (C.R.). Additionally, the use of commas vs. dots for decimals in p-values is inconsistent.

Solved

  • Model Interpretation:The SEM results (Figure 1 and Table 2) indicate that physical pain directly increases depression (beta=0.55) and decreases resilience (beta=-0.40), while resilience decreases depression (beta=-0.35). These findings are consistent with the literature and statistically robust.

thank you

  1. Discussion
  • Depth:The discussion relates findings to the literature but remains somewhat superficial. It fails to discuss the specific context of health science students—such as academic stress, clinical placement anxiety, or future career concerns—and how these factors might interact with pain and depression.

Solved. Line 189-197

  • References:The reference list contains a significant number of older sources (early 2000s) . Since this is a current topic, incorporating more research from the last 5-10 years would enhance the manuscript's relevance.

Dear reviewer. Right now, the references section lists 28 papers published between 2015 and 2025. Only 7 are from before 2015.

Conclusion and Recommendation

The study is worthy of publication due to its large sample size and robust statistical analysis (SEM). However, the typographical errors in tables (Critical Radio, SWEK) and text (depressión) give the impression of a hastily prepared manuscript. These errors must be corrected, and the discussion should be deepened to reflect the specific sample characteristics.

Solved

Best regards

Reviewer 3 Report

Comments and Suggestions for Authors

Dear Authors.

I am grateful for the opportunity to review your work. After reviewing and analysing your work, I would like to make a number of recommendations and suggestions. I would be grateful if you would take my comments into account as I believe that they can complete your manuscript.

Introduction

  • The introduction begins by providing data on physical pain in the U.S. and opioid consumption. However, these references are not very recent. Given that the study was conducted in Spain, it would be more appropriate for the data on chronic pain prevalence or morbidity and mortality to come from this country. National databases, such as the National Institute of Statistics, as well as other studies, could be useful for this purpose.
  • Regarding the hypothesis and objectives, these should be described continuously within the text, rather than in bullet points. Furthermore, they should be properly formulated, and information about the population in which the study will be conducted must be included.
  • When discussing depression, the manuscript describes sport in terms of its beneficial effects on depression. Since this is neither the objective nor a result of the study, it would be more relevant for the manuscript to include, for instance, information on depression among university students, as this is the reference population.

Metodology

  • It is mentioned that the study was conducted at several universities in Spain; however, it would be useful to specify which universities were involved.
  • Similarly, it is stated that the contact for completing the questionnaire was based on convenience, but through which medium was the online link sent? Were social media, the university's corporate email, etc., used?
  • The abstract mentions that the included universities were from nursing, physiotherapy, medicine, and sports sciences. However, the manuscript does not detail this selection. Was belonging to these academic fields an inclusion criterion? If not, could it be considered a limitation that the participants were from healthcare-related fields?
  • Figure 1 displays different font sizes between words. The rounded words are not displayed correctly. It would be useful to describe their meaning in the text and explain how they influence depression and resilience.

Results

  • In the data collection questionnaire, were sociodemographic and/or personal variables not included? In the results section, it would be interesting to include the first table describing these variables, and potential differences could be analysed based on the geographic location of the students, for example, in relation to depression.
  • In Table 1, the depression data (3.D) are shifted to the right, and it is not clear which column they belong to.

Discussion

  • Throughout the discussion, it would be beneficial to include a specific study that has evaluated these variables, particularly if it was conducted with university students. According to '...' et al. (X)…
  • Regarding the limitations, it would be pertinent to include the limitation related to the cross-sectional nature of the study, and then later mention the need for conducting longitudinal studies.

Best regards

Author Response

Reviewer

I appreciate your recommendations for publishing a higher quality paper.  I will highlight the changes you indicate in green. Thank you.

Introduction

  • The introduction begins by providing data on physical pain in the U.S. and opioid consumption. However, these references are not very recent. Given that the study was conducted in Spain, it would be more appropriate for the data on chronic pain prevalence or morbidity and mortality to come from this country. National databases, such as the National Institute of Statistics, as well as other studies, could be useful for this purpose.

 Solved Line 24-25

  • Regarding the hypothesis and objectives, these should be described continuously within the text, rather than in bullet points. Furthermore, they should be properly formulated, and information about the population in which the study will be conducted must be included.

 Solved Líne 78-81

  • When discussing depression, the manuscript describes sport in terms of its beneficial effects on depression. Since this is neither the objective nor a result of the study, it would be more relevant for the manuscript to include, for instance, information on depression among university students, as this is the reference population.

Solved Líne 57-66

Metodology

  • It is mentioned that the study was conducted at several universities in Spain; however, it would be useful to specify which universities were involved.

 Solved Líne 87-90

  • Similarly, it is stated that the contact for completing the questionnaire was based on convenience, but through which medium was the online link sent? Were social media, the university's corporate email, etc., used?

 Solved Líne 120

  • The abstract mentions that the included universities were from nursing, physiotherapy, medicine, and sports sciences. However, the manuscript does not detail this selection. Was belonging to these academic fields an inclusion criterion? If not, could it be considered a limitation that the participants were from healthcare-related fields?

 Solved Líne 249-251

  • Figure 1 displays different font sizes between words. The rounded words are not displayed correctly. It would be useful to describe their meaning in the text and explain how they influence depression and resilience.

 Solved Líne 143-145

Results

  • In the data collection questionnaire, were sociodemographic and/or personal variables not included? In the results section, it would be interesting to include the first table describing these variables, and potential differences could be analysed based on the geographic location of the students, for example, in relation to depression.

Dear reviewer, as structural equation modelling was used, the sociodemographic variables recorded were not taken into account. In previous studies published in the Healthcare journal, it was not necessary to include sociodemographic variables. Your proposal is very interesting, but in order to facilitate the publication of the article as soon as possible, we have decided not to include the variables you mention. Thank you. 

  • In Table 1, the depression data (3.D) are shifted to the right, and it is not clear which column they belong to.

Solved

Discussion

  • Throughout the discussion, it would be beneficial to include a specific study that has evaluated these variables, particularly if it was conducted with university students. According to '...' et al. (X)…

Solved Line 221-224 

  • Regarding the limitations, it would be pertinent to include the limitation related to the cross-sectional nature of the study, and then later mention the need for conducting longitudinal studies.

 Solved Line 240-241

Best regards

Round 2

Reviewer 1 Report

Comments and Suggestions for Authors

The authors have made all the comments and have contributed appropriately to the improvement of the manuscript

Author Response

The authors have made all the comments and have contributed appropriately to the improvement of the manuscript

Best Regard

Reviewer 3 Report

Comments and Suggestions for Authors

Thank you for taking my suggestions and comments into account. I would just like to point out:

  • The degrees of physiotherapy, nursing, medicine, and physical activity and sport sciences should be included in the design and participants (methodology), as they are in the abstract.
  • Table 1 is still slightly misaligned, and the first row appears to be shifted to the right, making it difficult to determine which data are located below.

Author Response

Dear reviewer,

I appreciate your recommendations for publishing a higher quality paper.  I will highlight the changes you indicate in blue. Thank you.

  • The degrees of physiotherapy, nursing, medicine, and physical activity and sport sciences should be included in the design and participants (methodology), as they are in the abstract.

Solved.Line 85-87

  • Table 1 is still slightly misaligned, and the first row appears to be shifted to the right, making it difficult to determine which data are located below.

I tried to improve the table as much as possible, taking your recommendation into account. I used the template provided by the journal. I tried to adjust the table within the possibilities offered by the template.

I am sending you a photo so that you can see how the table looks on my computer.  

 Best Regard
